# Impact of Mobility on Methicillin-Resistant *Staphylococcus aureus* among Injection Drug Users

**DOI:** 10.3390/antibiotics8020081

**Published:** 2019-06-17

**Authors:** Folashade B. Agusto, Soyeon Kim

**Affiliations:** 1Department of Ecology and Evolutionary Biology, University of Kansas, Lawrence, KS 66044, USA; 2Department of Molecular Biosciences, University of Kansas, Lawrence, KS 66044, USA; soyeonkim1199@ku.edu

**Keywords:** MRSA, injection drug users, mobility, sensitivity analysis, control strategies

## Abstract

In this study, we develop and present a deterministic model for the transmission dynamics of methicillin-resistant *staphylococcus aureus* (MRSA) among injection drug users. The model consists of non-injection drug users as well as low-and high-risk injection drug users (IDUs). The model further incorporates the movement of these individuals between large metro, suburban and rural areas. The model parameters were estimated by fitting the model to the 2008–2013 disease prevalence data for non-IDUs obtained from the Agency for Healthcare and Research and Quality (AHRQ), as well as the 2009–2013 Census Bureau data for the number of individuals migrating between three different counties in Kansas. Sensitivity analysis was implemented to determine the parameters with the most significant impact on the total number of infected individuals; the transmission probability, recovery rates, and positive behavioral change parameter for the subgroup have the most significant effect on the number of infected individuals. Furthermore, the sensitivity of the parameters in the different areas was the same when the areas are disconnected. When the areas are connected, the parameters in large-metro areas were the most sensitive, and the rural areas were least sensitive. The result shows that to effectively control the disease across the large metro, suburban and rural areas, it is best to focus on controlling both behavior and disease in the large metro area as this has a trickle-down effect to the other places. However, controlling behavior and disease at the same time in all the areas will lead to the elimination of the disease.

## 1. Introduction

Methicillin-resistant *staphylococcus aureus* (MRSA) is a bacteria resistant to antibiotics, particularly methicillin. It affects the subcutaneous tissue of the skin and often starts as a painful red bump that looks like a pimple or a spider bite. MRSA is difficult to treat due to its resistance to traditional antibiotics used in the treatment of Staph infections [1,2]; it can equally be deadly. The occurrence of MRSA started in 1960 just after two years of using it as a viable and effective treatment for methicillin-susceptible *staphylococcus aureus*.

MRSA is transmitted when a healthy individual comes in direct contact with an infected person with an open wound or an infected tissue or secretions. It can also be contracted indirectly from contaminated surfaces, objects, or materials contaminated with the bacteria by an infected person; the contaminated fomites can remain infectious for months [3]. A healthy individual becomes colonized with the bacteria until the MRSA bacteria finds an opportunity to infect the soft tissue of the skin [4].

Substance abuse, including drug abuse, is a cultural and public health problem affecting many millions of persons in the United States [5]. “Drug addiction is a complex health disorder characterized by its chronic and relapsing nature” [6]; it is preventable and treatable, and it is not due to any moral or societal failure or a criminal behavior [6].

MRSA is endemic among injection drug users (IDUs), its spread in the community started in 1981 and has since persisted in the community [4]. Some of the research studies on MRSA among IDUs are usually carried out by tracking the registered IDUs [7,8,9,10]. For example, Fleisch et al. [9] in their study followed 31 MRSA infected IDUs from August of 1994 to December 1999; they found that 19 of these individuals developed secondary, life-threatening ailments such as septic arthritis, endocarditis, pneumonia, and osteomyelitis. In another study, Binswanger et al. [8] found that 29 individuals displayed subcutaneous inflammation and infection from a study involving 169 IDUs.

Injection drug users present a unique set of behavioral factors that all accumulate to increase the risk of MRSA transmission and infection. For instance, Kretzschmar [10] found that 12% of the 494 IDUs in their study were HIV-positive while only 1.5% of 207 non-IDUs were HIV-positive and they concluded that HIV among these IDUs was not likely sexually transmitted [10]. In recent times, there has been an increase in the proportion of invasive MRSA cases among IDUs due to the use of non-sterile injection drug another risky behavior engaged by IDUs [11]. The number of IDUs increased from 4.1% in 2011 to 9.2% in 2016 [11]. Furthermore, IDUs are 16.3 times more likely to develop invasive MRSA infections than others [11].

Demographic studies of illicit drug use found a higher occurrence of illicit drug uses in large metropolitan populations [12]. The Department of Health and Human Services defined a large metropolitan area in their 2015 national survey of drug use and health as an area having a population greater than one million [13]. According to the Department, 8.3% of the individuals living in large metropolitan areas used drugs in the past month of the survey, followed by 8.2% of the populations living in small metropolitan areas with only 4.2% in rural areas [13]. In 2010 National Institute on Drug Abuse found very large cities (population over 500,000) have a higher lifetime prevalence of drug use at 64.9% of drug users followed by 63.7% in large cities (population of 100,000–500,000), 59.3% in medium cities [12]. According to the National Survey on Drug Use and Health (NSDUH) survey, the lifetime use of illicit drugs in large and small metropolitan areas was 113,967 while non-metropolitan areas were 16,644. These statistics bring to the light the great need for action in drug abuse interventions in cities, particularly in the metropolitan areas.

The goals in this paper are to determine the impact of movement on the control of MRSA when the communities are isolated and connected, and to also understand the effect of movement on drug rehabilitation programs and intervention as a means of controlling risky behaviors among IDUs when in isolated and connected communities. To achieve our goals, we extend the deterministic model in [14] to include multiple patches and movement between these patches.

This paper is organized as follows: In Section 2, we present the MRSA transmission model and carry out a theoretical study of its basic properties in Appendix A. In Section 2.2 we estimate the movement parameter values; values of the other model parameters are given in [14]. In Section 3, sensitivity analysis was carried out to identify the parameters that have significant impact on the model’s response function: The total number of infectious individuals in the community. In Section 4, we use the results obtained from the sensitivity analysis to investigate the impact of movement on both disease and behavioral controls in connected and isolated communities. The research discussions and conclusions are given in Section 5 and Section 6 respectively.

## 2. Model Formulation

To model the disease transmission, we consider MRSA transmission in three different areas namely large metropolitan, suburban, and rural areas. We developed a deterministic model for the transmission when these areas are connected. We use the deterministic model developed in [14] and extend it by including the movement of individuals between these connected areas. The methicillin-resistant *Staphylococcus aureus* transmission model among injection drug users divides individuals in the community into three subgroups, non-drug users, low-risk IDUs, and high-risk IDUs. The low-risk IDUs use drugs at a limited to moderate level, while high-risk IDUs are intense to severe drug users [14]. Each of these population subgroups is further divided into three compartments according to their disease status; namely uncolonized susceptible individuals ($U_{j}$), colonized individuals ($C_{j}$), and infected individuals ($I_{j}$), where $j=N_{i},L_{i},H_{i}$ for non-injection drug users, low-risk IDUs, and high-risk IDUs and i=M,S,R for large metropolitan, suburban, and rural areas, respectively. These lead to the following total population.
$$N_{i}=U_{N_{i}}+C_{N_{i}}+I_{N_{i}}+U_{L_{i}}+C_{L_{i}}+I_{L_{i}}+U_{H_{i}}+C_{H_{i}}+I_{H_{i}},$$

The MRSA transmission dynamics with IDUs and mobility within three different areas is given by the following system of ordinary differential equations
(1)$$\begin{array}{ccc} \frac{dU_{N_{i}}}{dt} & = & \pi_{N_{i}}+\alpha_{L_{i}}U_{L_{i}}+\tau_{N_{i}}C_{N_{i}}+\gamma_{N_{i}}I_{N_{i}}-\lambda_{N_{i}}U_{N_{i}}-k_{1_{i}}U_{N_{i}}-\sum_{j=S,\;j\neq i}^{R}\psi_{ij}U_{N_{i}}+\sum_{j=S,\;j\neq i}^{R}\psi_{ji}U_{N_{i}}\\ \frac{dC_{N_{i}}}{dt} & = & \alpha_{Li}C_{Li}+\lambda_{Ni}U_{Ni}-\tau_{Ni}C_{Ni}-k_{2_{i}}C_{Ni}-\sum_{j=S,\;j\neq i}^{R}\psi_{ij}C_{N_{i}}+\sum_{j=S,\;j\neq i}^{R}\psi_{ji}C_{N_{i}}\\ \frac{dI_{N_{i}}}{dt} & = & \alpha_{Li}I_{Li}+\sigma_{Ni}C_{Ni}-k_{3_{i}}I_{Ni}-\sum_{j=S,\;j\neq i}^{R}\psi_{ij}I_{N_{i}}+\sum_{j=S,\;j\neq i}^{R}\psi_{ji}I_{N_{i}}\\ \frac{dU_{L_{i}}}{dt} & = & \pi_{Li}+\omega_{Ni}U_{Ni}+\alpha_{Hi}U_{Hi}+\tau_{Li}C_{Li}+\gamma_{Li}I_{Li}-\lambda_{Li}U_{Li}-k_{4_{i}}U_{L_{i}}-\sum_{j=S,\;j\neq i}^{R}\psi_{ij}U_{L_{i}}+\sum_{j=S,\;j\neq i}^{R}\psi_{ji}U_{L_{i}}\\ \frac{dC_{L_{i}}}{dt} & = & \omega_{Ni}C_{Ni}+\alpha_{Hi}C_{Hi}+\lambda_{Li}U_{Li}-\tau_{Li}C_{Li}-k_{5_{i}}C_{L_{i}}-\sum_{j=S,\;j\neq i}^{R}\psi_{ij}C_{L_{i}}+\sum_{j=S,\;j\neq i}^{R}\psi_{ji}C_{L_{i}}\\ \frac{dI_{L_{i}}}{dt} & = & \omega_{Ni}I_{Ni}+\alpha_{Hi}I_{Hi}+\sigma_{Li}C_{Li}-k_{6_{i}}I_{Li}-\sum_{j=S,\;j\neq i}^{R}\psi_{ij}I_{L_{i}}+\sum_{j=S,\;j\neq i}^{R}\psi_{ji}I_{L_{i}} \end{array}$$
(2)$$\begin{array}{ccc} \frac{dU_{H_{i}}}{dt} & = & \pi_{H_{i}}+\omega_{L_{i}}U_{L_{i}}+\tau_{H_{i}}C_{H_{i}}+\gamma_{H_{i}}I_{H_{i}}-\lambda_{H_{i}}U_{H_{i}}-k_{7_{i}}U_{H_{i}}-\sum_{j=S,\;j\neq i}^{R}\psi_{ij}U_{H_{i}}+\sum_{j=S,\;j\neq i}^{R}\psi_{ji}U_{H_{i}}\\ \frac{dC_{H_{i}}}{dt} & = & \omega_{L_{i}}C_{L_{i}}+\lambda_{H_{i}}U_{H_{i}}-\tau_{H_{i}}C_{H_{i}}-k_{8_{i}}C_{H_{i}}-\sum_{j=S,\;j\neq i}^{R}\psi_{ij}C_{H_{i}}+\sum_{j=S,\;j\neq i}^{R}\psi_{ji}C_{H_{i}}\\ \frac{dI_{H_{i}}}{dt} & = & \omega_{L_{i}}I_{L_{i}}+\sigma_{H_{i}}C_{H_{i}}-k_{9_{i}}I_{H_{i}}-\sum_{j=S,\;j\neq i}^{R}\psi_{ij}I_{H_{i}}+\sum_{j=S,\;j\neq i}^{R}\psi_{ji}I_{H_{i}} \end{array}$$
where
$$\begin{array}{ccc} k_{1_{i}} & = & \omega_{N_{i}}+\mu_{i},\;\;k_{2_{i}}\;=\;\omega_{N_{i}}+\sigma_{N_{i}}+\mu_{i},\;\;k_{3_{i}}\;=\;\omega_{N_{i}}+\gamma_{N_{i}}+\mu_{i}+\delta_{N_{i}}\\ k_{4_{i}} & = & \omega_{L_{i}}+\alpha_{L_{i}}+\mu_{i},\;\;k_{5_{i}}\;=\;\omega_{L_{i}}+\alpha_{L_{i}}+\sigma_{L_{i}}+\mu_{i},\;\;k_{6_{i}}\;=\;\omega_{L_{i}}+\alpha_{L_{i}}+\gamma_{L_{i}}+\mu_{i}+\delta_{L_{i}}\\ k_{7_{i}} & = & \alpha_{H_{i}}+\mu_{i},\;\;k_{8_{i}}\;=\;\alpha_{H_{i}}+\sigma_{H_{i}}+\mu_{i},\;\;k_{9_{i}}\;=\;\alpha_{H_{i}}+\gamma_{H_{i}}+\mu_{i}+\delta_{H_{i}}. \end{array}$$
and i=M,S,R for the different areas, i.e., large metropolitan, suburban, and rural communities. The flow diagram for the model is depicted in Figure 1 for a single patch *i*. The associated model variables and parameters are described in Table 1.

The parameter $\pi_{N_{i}}$ is the recruitment rate into the uncolonized non-injection drug users ($U_{N_{i}}$) subgroup for a particular area or patch *i*. The parameter μ is the natural death rate in each subpopulation in patch *i*. The force of infection of the non-injection drug users in patch *i* is given by
$$\begin{array}{ccc} \lambda_{X_{i}} & = & \frac{\beta_{X_{i}}\left(C_{N_{i}}+I_{N_{i}}+C_{L_{i}}+I_{L_{i}}+C_{H_{i}}+I_{H_{i}}\right)}{N_{i}} \end{array}$$
where, X=N,L,H for the different subgroups according to their behavior (i.e., non IDUs, low-and high-risk IDUs) and the parameter $\beta_{N_{i}}$ is the probability in patch *i* of an uncolonized individual becoming colonized following exposure to the MRSA bacteria through contact with colonized and infected individuals in either the non-injection drug users class or the low-risk class or the high-risk class. Colonized individuals in patch *i* are decontaminated at the rate $\tau_{N_{i}}$ [15], after which they move back to the uncolonized class. However, as the disease progresses the colonized non-drug users ($C_{N_{i}}$) in patch *i* move to the infected class at rate $\sigma_{N_{i}}$. Infected individuals ($I_{N_{i}}$) recover at the rate $\gamma_{N_{i}}$ and thus move to the uncolonize class ($U_{N_{i}}$). The model does not include recovered compartment since there is no immunity associated with the infection. The parameters and transitions for the low-risk and high-risk populations in patch *i* are similarly defined (with the subscript $N_{i}$ replaced by $L_{i}$ and $H_{i}$ respectively).

If non-drug users in patch *i* engage in risky behavior resulting in injection drug use, we assume that these individuals will initially start small with less frequent and less risky behavior and drug use; therefore, these individuals will move into the low-risk IDU subgroup from the non-drug user subgroup at rate $\omega_{N_{i}}$. However, these individuals can decrease these risky behaviors either by timely family interventions or by enrolling in drug rehabilitation programs [7,16] or by other kinds of interventions; when this happens, they go back to the non-drug user subgroup at the rate $\alpha_{L_{i}}$.

Individuals in the low-risk subgroup in patch *i* might progress and become high-risk injection drug users due to their persistent engagement in these risky behaviors, thereby moving into the high-risk IDUs subgroup at the rate $\omega_{L_{i}}$. As with the low-risk individual, these individuals may stop injecting drugs when they enter a drug rehabilitation program or utilizing needle exchange programs [7,16]. We assume these disease and behavioral processes are not instantaneous but rather a gradual process; hence these individuals due to their behavior first enter the low-risk subgroup at rate $\alpha_{H_{i}}$.

Note that patch *i* refer to either large metropolitan, suburban or rural areas. We will investigate the effect of movement on both behavior and disease controls when these patches are either connected or isolated.

### 2.1. Reproductive Number $R_{0}$

The basic reproductive number, $R_{0}$, is given by the result of the expression
$$\begin{array}{ccc} R_{0} & = & \rho (FV^{-1}). \end{array}$$
where ρ is the spectral radius.

The basic reproductive number, $R_{0}$, is the average number of secondary infections produced by one infectious individual introduced into a wholly susceptible population [17,18,19,20]. If $R_{0}<1$, MRSA will die out in the community, and if $R_{0}>1$, MRSA it will persist and become endemic. The epidemiological implication of this result is that it will be possible to eliminate MRSA from the community provided the reproduction number ($R_{0}$) is brought to (and maintained at) value less than one.

The theoretical study of the basic quantitative properties of the model are presented in Section A.1, and the derivation of the expression of basic reproductive number, $R_{0}$, are given in Section A.2.

### 2.2. Model Parameter Estimation

#### 2.2.1. MRSA Demographic Data from 2008–2013

To parameterize the MRSA model in [14], we used the 2008–2013 demographic data obtained from the Agency for Healthcare and Research and Quality (AHRQ) [21]. AHRQ compiles discharge data from hospitals throughout the United States; this data is classified as the International Classification of Diseases (ICD-9). Any patients discharged with the MRSA are recorded with ICD-9 code. To obtain the MRSA data from AHRQ discharge data, we set the demographic targets to look at large metro, large suburb, and rural areas. The large metropolitan population produced the highest number of patients with MRSA listed on their medical records upon discharge. Figure 2 displays the data in tens of thousands of hospital discharges.

#### 2.2.2. Parameter Estimation

The parameters in the single patch model in [14] of the MRSA model Equation (Equation 1) was obtained using a reduced model involving only the non-IDUs. The ICD-9 MRSA data for large metro, rural, and suburb regional areas was used to estimate the parameters. The results are given in Table 2; the model simulation profile and the fitted data are depicted in Figure 3.

To obtain the parameter estimates for the IDUs subgroups in the full MRSA model Equation (Equation 1) for a single patch, we follow the approach in [14] and multiplied the non-IDUs subgroup parameter estimates $\beta_{N_{i}},\gamma_{N_{i}},$ and $\sigma_{N_{i}}$ by modifying parameters $\varepsilon_{L}$ and $\varepsilon_{H}$ representing low-risk and high-risk subgroups. These risk factors (modification parameters) listed in Table 3 enable us to determine an estimate away (either in an increasing or decreasing form) from the non-IDUs parameters. Their derivation is given in [14]. It is possible to also estimate the risk factors using addiction severity score [22]. We assume that the values of the parameters in each subgroup are the same for the three different areas we are considering.

#### 2.2.3. Estimation of the Movement Rates $\psi_{ij}$ and $\psi_{ji}$

To estimate the movement rates $\psi_{ij}$ and $\psi_{ji}$, we focus on the number of people migrating between three counties in Kansas, namely Shawnee, Douglass, and Jefferson counties. These counties were chosen to represent the Large metropolitan, suburban and rural areas. Although the sizes of these counties are smaller than the size of the metropolitan area defined by the Department of Health and Human Services in Section 1 which puts the size of a large metro as a million. Nevertheless, the result and conclusion are not expected to deviate from the result obtained if we have used a larger population size. The population in these counties (Shawnee, Douglass, and Jefferson counties) from 2009–2013 according to the Census Bureau 2009–2013 data are 175,136, 110,973, and or 18,947, respectively. This period was used to match with the time range of the MRSA discharged data used to estimate the disease parameters. We have not included 2008 in this time range because the Census Bureau estimate of the gross number migrating for 2008 overlaps with other time range and cannot be tease apart.

Nevertheless, we believe the 2009–2013 estimate is a good estimate for the entire 2008–2013 time range. The number of people migrating between these counties over this period was obtained from the county-to-county 2009–2013 ins-outs-nets-gross migration data from the Census Bureau [23]. Table 4 shows the number of individual migrating between each county. The migration rate is obtained by dividing the number of individuals migrating by the total population in each county, see Table 5 and Figure 4. Figure 4 shows more people moving to the Shawnee county from Douglas, and Jefferson counties; furthermore, more people move to Shawnee than Douglas from Jefferson. However, fewer people migrate in the opposite direction from the larger counties to Jefferson.

.

## 3. Sensitivity Analysis

Outputs from deterministic models are governed by their input parameters, which often exhibit some level of uncertainty in the process of their selection. Exploration of the MRSA model Equation (Equation 1) above requires the identification of parameters which significantly affect the number of infected individuals (the key model outputs). To assess the impact of the input parameters uncertainty and the sensitivity of the key model outputs from the numerical simulations to variations in each parameter, we carried out a global sensitivity analysis using Latin Hypercube Sampling (LHS) and partial rank correlation coefficients (PRCC).

The LHS is a stratified sampling technique without replacement; it allows for an efficient analysis of parameter variations across simultaneous uncertainty ranges in each parameter [24,25,26,27]. The PRCC method measures the strength of the relationship between the parameters and the model outcome. It gives the degree to which each parameter affects the model outcome [24,25,26,27].

To generate the LHS matrices, we assume a uniform distribution for all the model parameters. We then carried out a total of 1000 simulations (runs) of the models for the LHS matrix, using the parameter values given in Table 2 and Table 3 (with ranges varying from ±20% the stated baseline values) and as response functions, the sum of carriers and infected in each region for the isolated case; for the connected case, the response function is the sum of the carriers and infected across the regions. These simulation runs were followed by the parameter ranking using PRCC.

The most significant parameters with the largest impact on the model outcome examined under each analysis have the largest partial rank correlation coefficient (PRCC) values. The parameters with positive PRCC values have a positive impact on the number of infected and increase in their values will lead to an increase in the number of infected. While the parameters with negative PRCC values will reduce the number of infected due to their negative impact on the carriers and infected individuals in the population.

The key parameters with negative PRCC values when the regions are isolated are $\alpha_{HM},\;\gamma_{HM},$$\;\gamma_{LM},\gamma_{NM},\;\alpha_{LM},\;\gamma_{HS},\;\gamma_{LS}$ and $\alpha_{HS}$. While those with positive PRCC values are $\sigma_{HM},\;\beta_{HM},$$\;\beta_{LM},\beta_{NM},\;\beta_{HS},\;\beta_{LS},\;\omega_{HM},\;\omega_{LM}$, and $\omega_{LS}$. It is interesting to note that the PRCC values are the same in all the patches when the patches are in isolation, see Figure 5.

Figure 6 depict the PRCC values when the regions are connected. The key parameters with negative PRCC values are $\alpha_{HM},\;\gamma_{HM},\;\gamma_{LM},\;\gamma_{NM},\;\alpha_{LM},\;\gamma_{HS},\;\gamma_{LS}$ and $\alpha_{HS}$. While those with positive PRCC values are $\sigma_{HM},\;\beta_{HM},\;\beta_{LM},\;\beta_{NM},\;\beta_{HS},\;\beta_{LS},\;\omega_{HM},\;\omega_{LM}$, and $\omega_{LS}$. Under this case, the PRCC values are higher in the large metro area followed by the suburban area, the rural area have the lease values. This is due to the high movement rates into the large metro areas compare to the other two areas.

## 4. Control Measures

In this section we considered two types of control strategies; the first strategy in Section 4.1 investigates the impact of controlling an individual’s risky behaviors which promotes injection drug use via the vertical downward movement parameters $\omega_{N}$ and $\omega_{L}$, and the enrollment in rehabilitation programs via the vertical upward movement parameters $\alpha_{L}$ and $\alpha_{H}$. The second strategy in Section 4.2 investigates the control of the disease in the community using results of the highly effective strategy identified in [14]. For the two different control strategies either control of risky behavior or disease, we used the intermediate risk factor values $\varepsilon_{L}=21$% and $\varepsilon_{H}=66$ % as was done in [14].

### 4.1. Control of Risky Injection Drug Use Behaviors

In this section, we investigate the impact of behavioral change due to vertical downward and upward movements between the sub-groups from risky behaviors that promote (upward movement) and discourages (downward movement) injection drug use in the community. Note, our focus here is to control the risky behaviors and not MRSA transmission within the subgroups of the different areas.

We set the baseline behavioral downward and upward movements parameters as $\omega_{N}=0.05825$, $\omega_{L}=0.116$, $\alpha_{L}=0.0112$, $\alpha_{H}=0.0560$. Since we are interest in investigating the effect of behavioral control, on movement between the patches, we decrease the risky behavior by 2 and double the positive behavior. Thus, we set the behavioral parameters as $\omega_{N}=0.05825/2$, $\omega_{L}=0.116/2$, $\alpha_{L}=0.0112\times 2$, $\alpha_{H}=0.0560\times 2$.

#### 4.1.1. Behavioral Control When Patches Are Isolated

We start by considering the case where the patches are in isolation. We observe in Figure 7 a reduction in the number of cases in each of the patches. We have normalized the number infected for proper comparison between the patches. We observed further from the figure an equal number of infected individuals regardless of the patch size. Table 6 gives the proportion of infected individuals at the end of the simulation period.

#### 4.1.2. Behavioral Control in a Patch One at a Time When the Patches are Connected

Under this scenario, the patches are connected, and behavioral control is implemented in a patch one at a time. By behavioral control, we mean discouraging risky behavior and encouraging IDUs to either practice safe needle use or to enter rehabilitation centers. In Figure 8a–c, the control is in the Large metro area only, and we observed a reduction in the proportion of infected individuals; however, this number is more than when the patch is isolated (compare Table 6 and Table 7). In addion, we observed that controlling behavior in large metropolitan area leads to a decrease to some extent in the number of infected individuals in the suburban and rural. However, the effect is stronger in the rural area than the suburban. We observed similar dynamics in Figure 8d–f when we are controlling behavior in the suburban patch. This, however, is not the case when we control behavior in the rural area; we see a negligible effect in the patches with large population even though more people move into these patches from the rural area, see Figure 8g–i, Table 7 gives the proportion of infected at the end of the simulation period.

The negligible effect of the control might be due to the fact that the number of individuals coming in from the rural area to the larger populated areas (large metropolitan and suburban areas) is small in comparison to the overall population size in the larger areas, even though the rate from rural area is higher (see Figure 4). As a result, their overall influence might be weak. This study have not looked at the effect of super-spreaders or influencers, we have assumed that every individual in the community have the same global effect. This will be an interesting future work to consider.

### 4.2. Control of MRSA Transmission

In this section, we investigate the impact of controlling MRSA transmission within the subgroups when the patches are isolated and when they are connected. To do this, we use results obtained from the sensitivity analysis (see also [14]). For the sensitivity analysis we obtained control strategies that reduce $\beta_{N_{i}}$ and increase $\gamma_{N_{i}}$, i=M,S,R, will impact MRSA in the community, since these parameters have strong positive ($\beta_{N_{i}}$) and negative ($\gamma_{N_{i}}$) impact on the number of infected; the same holds for the parameters $\beta_{L_{i}}$, $\beta_{H_{i}}$, $\gamma_{L_{i}}$, and $\gamma_{H_{i}}$. In [14], three different strategies were implemented namely low-effectiveness strategy, moderate-effectiveness strategy, and high-effectiveness strategy and it was concluded in [14] that the high-effectiveness strategy did the best in reducing infection in the community. In this study, we focus on the high-effectiveness strategy and set in each patch the following disease-related parameters as

$$\begin{array}{ccc} \beta_{N_{i}} & = & \beta_{N_{i}}/4,\;\;\;\;\;\;\;\;\;\;\;\;\gamma_{N_{i}}\;=\;\left(1+0.009\right)\gamma_{N_{i}},\;\;\;\;i=M,S,R\\ \beta_{L_{i}} & = & \beta_{N_{i}}\left(1+\varepsilon_{L}\right),\;\;\;\;\beta_{H_{i}}\;=\;\beta_{N_{i}}\left(1+\varepsilon_{H}\right)\\ \gamma_{L_{i}} & = & \gamma_{N_{i}}\left(1-\varepsilon_{L}\right),\;\;\;\;\gamma_{H_{i}}\;=\;\gamma_{N_{i}}\left(1-\varepsilon_{H}\right). \end{array}$$

#### 4.2.1. MRSA Control When the Patches Are Isolated

Considering the scenario when the patches are in isolation, we observe a substantial reduction in the number of cases in each of the patches (see Figure 9 and Table 7). The level of reduction is the same in each patch regardless of the patch size, and it’s more than when controlling behavior in the isolated patches case, see Table 6 and Table 7 for the proportion infected at the end of the simulation period.

#### 4.2.2. MRSA Control in a Patch One at a Time When the Patches Are Connected

Next, we consider the case where the disease is controlled one at a time when the patches are connected. We see in Figure 10a substantial reduction in infection level in the individual patch where the control is being implemented. The reduction is not as much as what we see when they are isolated (compare Table 8 and Table 9). Furthermore, the effect of the control in either the metro or suburban area is negligible when the control is in either place and slightly more when in the rural area, see Table 9. For instance, when the control is implemented in the large metro area, we see a smaller reduction in the number of infected in the suburban area compare to the number in the large metro and rural areas. This is because the rate at which people are moving to the metro from the suburban is higher than the rate at which they are moving from the metro to the suburban see Figure 4, and their population (metro and suburban) is relatively comparable in size. In addition, the rate at which people move to the metro from the rural area is higher than the rate from the metro to the rural; when converted to numbers, we see that more people actually move from the metro to the rural area than those that move from the rural area to the metro; this is the reason why the effect of the control in large metropolitan area is felt more in the rural area. In addition, the control of the disease in the rural area have no effect in the larger patches.

### 4.3. MRSA and Behavioral Controls

In this section, we investigate the effect of controlling both behavior and disease transmission at the same time when the patches are isolated and when they are connected. We set the behavioral parameters as $\omega_{N}=0.05825/2$, $\omega_{L}=0.116/2$, $\alpha_{L}=0.0112\times 2$, $\alpha_{H}=0.0560\times 2$, and the disease parameters as

$$\begin{array}{ccc} \beta_{N_{i}} & = & \beta_{N_{i}}/4,\;\;\;\;\;\;\;\;\;\;\;\;\gamma_{N_{i}}\;=\;\left(1+0.009\right)\gamma_{N_{i}}\\ \beta_{L_{i}} & = & \beta_{N_{i}}\left(1+\varepsilon_{L}\right),\;\;\;\;\beta_{H_{i}}\;=\;\beta_{N_{i}}\left(1+\varepsilon_{H}\right)\\ \gamma_{L_{i}} & = & \gamma_{N_{i}}\left(1-\varepsilon_{L}\right),\;\;\;\;\gamma_{H_{i}}\;=\;\gamma_{N_{i}}\left(1-\varepsilon_{H}\right). \end{array}$$

#### 4.3.1. MRSA and Behavioral Control in Isolated Patches

We start with the case when the patches are isolated, and we are controlling both behavior and disease transmission. We see in Figure 11 that the disease is just about eliminated, with very low infection levels. This, of course, happens in all the three isolated patches.

#### 4.3.2. MRSA and Behavioral Control in a Patch One at a Time in Connected Patches

Next, we consider the case when the patches are connected, and both behavior and disease are being controlled in the patches one at a time. We observed under this scenario a significant reduction in the number of infected in both large metropolitan and suburban when control is implemented in either place. The proportion of infected is more than what was obtained when the patches are in isolation. Their effect on each other (large metro and suburban areas) is not as strong as with isolated patches. This effect carries through to the rural area as well with the control in the large metropolitan area having a stronger effect than the control in the suburban area, see Figure 12 and Table 10.

On the other hand, when the control is implemented in the rural area, the effect on the large metro and suburban areas is negligible. Although we are able to reduce the number of infection in the rural area under this strategy; the reduction is not as low as what was obtained under the isolated case.

### 4.4. MRSA and Behavioral Control in Connected Patches

Lastly, under this scenario, we implement both behavior and disease control when the region is connected, and we see in Figure 13 an elimination of the disease across the three patches.

## 5. Discussion

In this study, we extended the model developed in [14] to include the movement of individuals between different patches (metropolitan, suburban, and rural areas). The model further includes the movement of individuals within different subgroups into different patches due to behavioral changes. The goal of the study is to determine how such mobility of individuals within the different patches will affect interventions either those targeting individuals behavior or those aimed at curtailing the disease transmission and spread. Note, that we have not aimed to study the impact of movement on disease spread.

The result shows that when we control risky behaviors either by the use of programs like the clean needle exchange or encourage individuals to enter drug rehabilitation programs, we are able to reduce the disease but not eliminate the disease regardless of the connections between the regions. We see a substantial reduction in isolated patches unlike when the patches are connected, and we were doing a targeted control in one area at a time. These results agree with existing research [28,29,30] which note that only a 25% to 40% partial reduction of HIV transmission risk among IDUs can be obtained from interventions which focus solely on individual behavioral change. However, some combination of individually oriented interventions may further reduce the incidence of HIV among IDUs but not completely eliminate it [30,31].

Mobility of individuals plays a vital role in the spread of communicable disease. The movement of individuals to an area increases the likelihood of disease occurrence. Population movement and mixing is one of the critical factors identified in the social structural production of HIV risk associated with drug injecting behaviors [30]. Furthermore, movement of IDUs within social network significantly increases the likelihood of their risky injection drug use behavior over time [32]. For instance, transient individuals due to homelessness play a role in the spread of HIV among IDUs as these individuals were more likely to engage in risky behaviors and sharing of needles and going to a shooting gallery than non-transient individuals [33]. These residential transients often have more than one sleeping locations and a higher chance of syringe sharing [34].

Hence, it is not surprising to see the results from our sensitivity analysis which showed high sensitivity indices for parameters associated with the large metropolitan areas which have a high flow of individuals into it. Although, our model assumes that all individuals from one local to another have the same mobility rates regardless of their behavior. Recall that the large metropolitan area has a large number of people compared to the rural area. This result implicitly aligns with Hoffmann et al. [32], even though we have used a deterministic model unlike the network model used by Hoffmann et al.

The mobility of injection drug users is a key mechanism for the introduction of both IDUs and infectious diseases into low prevalence areas [35,36,37]. The disease flow is usually from urban centers into rural areas; these rural localities often lack adequate disease prevention and harm reduction services [35,38,39]. Figure 4 shows the migration rate from the larger Shawnee county to the smaller Jefferson county, a county with fewer residents. Although the rural to urban migration has a higher flow of migrating individuals, we still see a flow from the larger county to the smaller county which will impact the distribution of the disease in the smaller county.

Accessing the effect of the intervention on MRSA transmission in our current study shows a considerable reduction in infection level when each patch implements disease control strategies, see Figure 10. Since there is this backward flow of individuals into the rural area from the larger populated metropolitan and suburban areas, it is not surprising to see the effect of this control trickling into the rural area leading to a noticeable reduction in the number of infection in the rural areas as risky behavior is being controlled in a large metropolitan and suburban areas. Even though these rural communities do not have infrastructures to deal with the disease spread and rehabilitation [35,38,39]. From our study, we see that the effect of the control in the rural area is hardly felt in the large metro and suburban areas. This might be due to the overall people migrating. However, we see a stronger effect in the rural area when both control strategies (i.e., control behavior and disease) are implemented in one area at a time; first in the metropolitan area then in the suburban area, see Figure 12. The effect in the metro area is stronger than the suburban area.

Thus, top-down rather than the bottom-up approach in targeting MRSA in the community with IDUs is a better strategy to controlling the disease in rural communities that might be struggling with resources or are in hard to reach places. As the effect of the treatment either behavioral or disease control will trickle down to surrounding smaller communities from larger communities implementing such controls. The same cannot be said for the reverse bottom-up approach as the effect of the interventions is only local to the rural area.

An underlining assumption in this study is that individuals have stable housing structures once they migrate from one locale to the other. As earlier noted, homeless transients often have more than one sleeping locations which might increase and encourage needle sharing practices and behaviors. In future studies, we will investigate the effect of transient mobility due to homelessness and the impact of various interventions on disease transmission and spread.

## 6. Conclusions

In this study, we investigated the effect of the mobility of individuals within different regions on control interventions either those targeting individuals behavior or those aimed at curtailing the disease transmission. We used an extended version of the model developed in [14] to include the movement of individuals between different regions. The model also includes the movement of individuals within different subgroups into different regions. We itemize our results below.
(1)Sensitivity analysis using LHS and PRCC shows that
(i)Isolated communities have the same sensitivity index;(ii)Connected communities have different sensitivity index, the large metro area with the highest movement rates have the highest sensitivity index;(2)The parameter with the highest impact are related to positive change in behavior ($\alpha_{Xi}$), transmission probability ($\beta_{Xi}$), and recovery ($\gamma_{xi}$) rates in each group of each connected communities;(3)In controlling risky behaviors and MRSA among IDUs we observed that
(i)Controlling risky behaviors or MRSA in isolated areas can substantially reduce MRSA in the communities;(ii)Controlling risky behavior or MRSA in the large metro area trickles to other connected areas,(iii)Interventions implemented in the large metropolitan areas have the strongest impact on the disease spread followed by suburban area;(iv)Inventions in the rural area have no effect on the other connected areas;(4)The disease can be eliminated if both MRSA and behavior are controlled in the communities whether isolated or connected

## Figures and Tables

**Figure 1 antibiotics-08-00081-f001:**
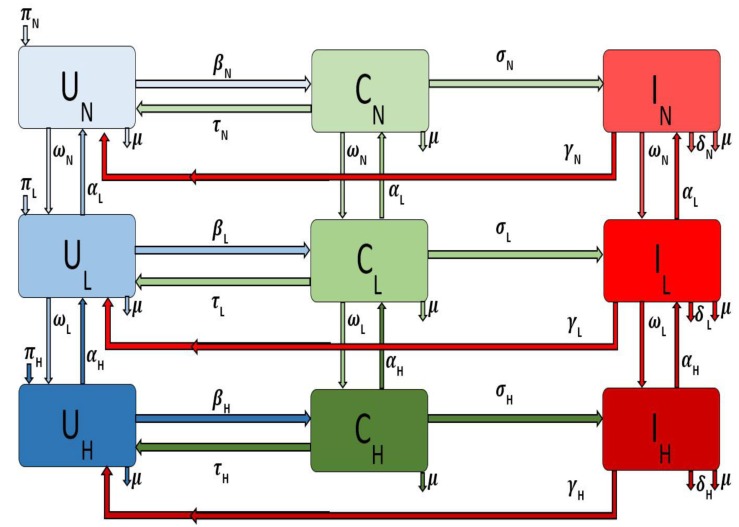
Flow diagram of the Methicillin-resistant *staphylococcus aureus* (MRSA) model Equation (Equation 1) with injection drug users (IDUs) in a single patch.

**Figure 2 antibiotics-08-00081-f002:**
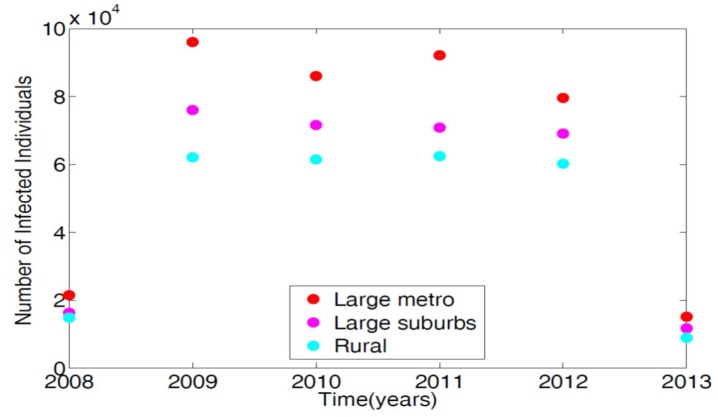
Demographic data from three regional areas, large metro, large suburb, and rural. Large metropolitan data gives the highest number of MRSA patients discharged with MRSA ICD-9 code listed on their medical record in the magnitude of tens of thousands.

**Figure 3 antibiotics-08-00081-f003:**
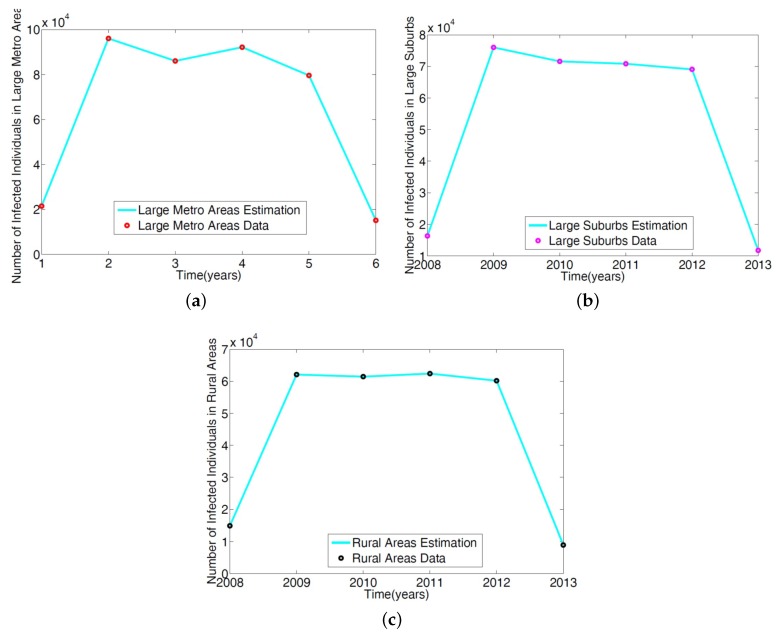
Simulation of the non-IDUs reduced MRSA model given in Wagner [14] was fitted to the ICD-9 MRSA data for: (**a**) Large metropolitan data; (**b**) large suburbs area data; (**c**) rural area data.

**Figure 4 antibiotics-08-00081-f004:**
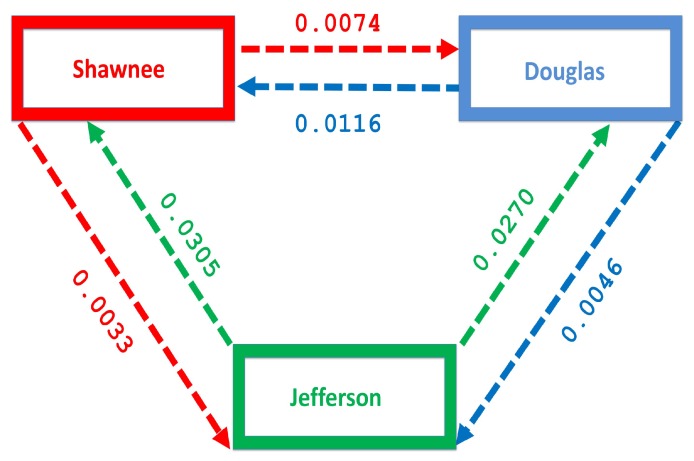
Movement rates within the regions.

**Figure 5 antibiotics-08-00081-f005:**
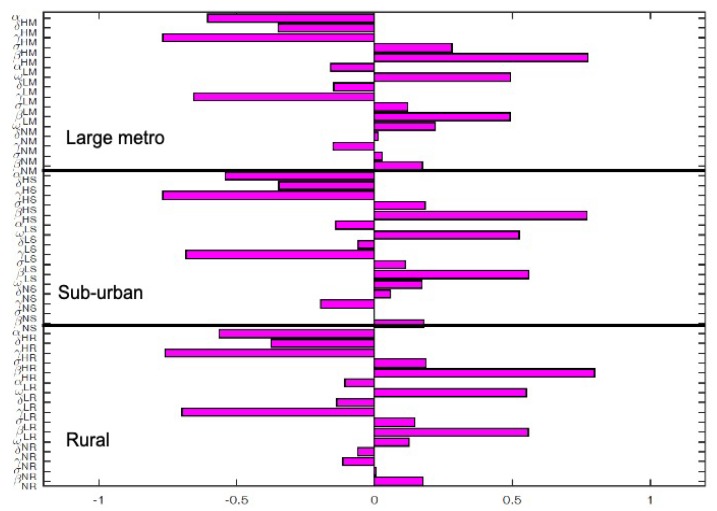
Partial rank correlation coefficient (PRCC) values of the MRSA model using as response function infected individuals when the patches (large metropolitan, suburban, and rural areas) are isolated. Parameter values used are given in Table 2, Table 3 and Table 4.

**Figure 6 antibiotics-08-00081-f006:**
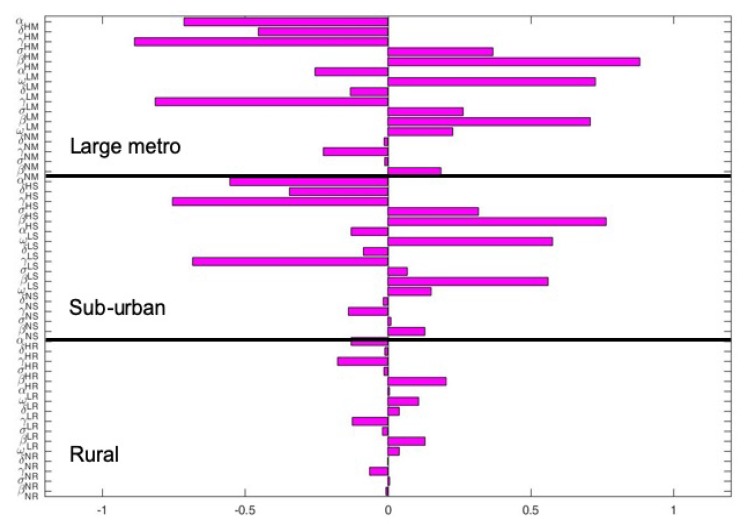
Partial rank correlation coefficient (PRCC) values of the MRSA model using as response function infected individuals when the regions (metro, suburban, and rural areas) are connected. Parameter values used are given in Table 2, Table 3 and Table 4.

**Figure 7 antibiotics-08-00081-f007:**
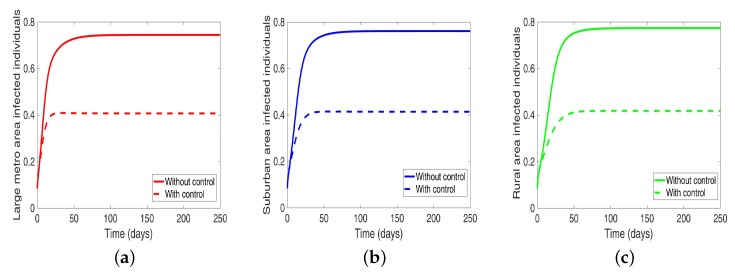
MRSA model simulation with no movement. (**a**) Infected individuals in large metro area, (**b**) Infected individuals in suburban area. (**c**) Infected individuals in rural area. Parameter values used are given in Table 2 and Table 3.

**Figure 8 antibiotics-08-00081-f008:**
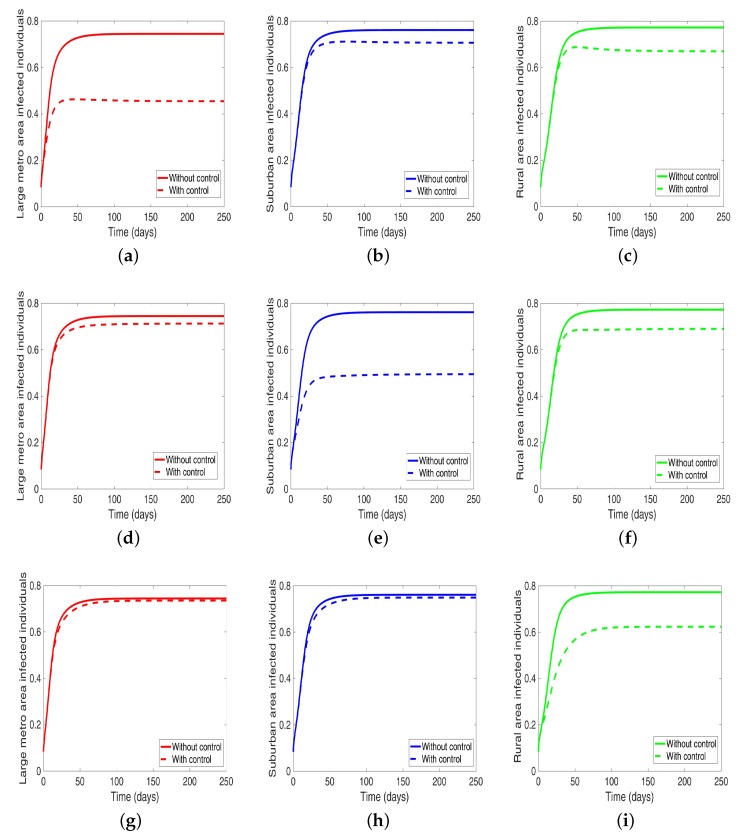
MRSA model simulation with movement and behavioral control in large metro area only (**a**–**c**). Movement and behavioral control in suburban area only (**d**–**f**). Movement and behavioral control in rural area only (**g**–**i**). (**a**) Infected individuals in large metro area. (**b**) Infected individuals in suburban area. (**c**) Infected individuals in rural area. (**d**) Infected individuals in large metro area. (**e**) Infected individuals in suburban area. (**f**) Infected individuals in rural area,=. (**g**) Infected individuals in large metro area. (**h**) Infected individuals in suburban area. (**i**) Infected individuals in rural area. Parameter values used are given in Table 2, Table 3 and Table 4.

**Figure 9 antibiotics-08-00081-f009:**
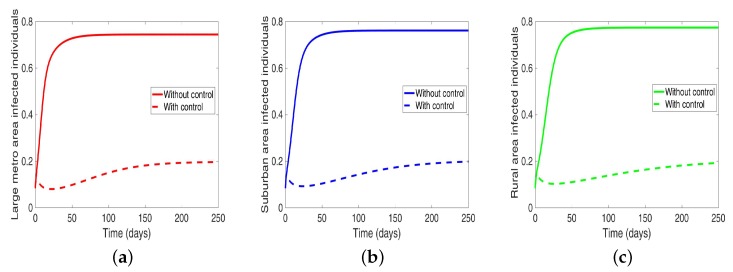
MRSA model simulation with no movement. (**a**) Infected individuals in large metro area, (**b**) Infected individuals in suburban area. (**c**) Infected individuals in rural area. Parameter values used are given in Table 2, Table 3 and Table 4.

**Figure 10 antibiotics-08-00081-f010:**
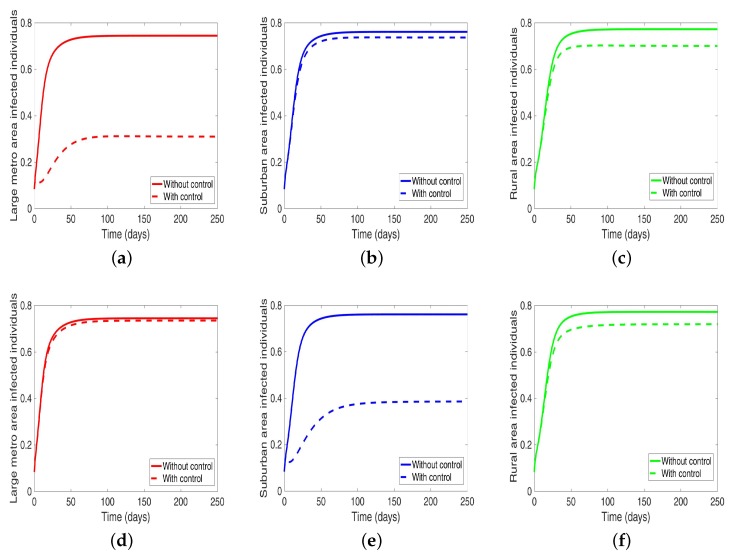
MRSA model simulation with movement, disease, and behavioral control in large metro area only (**a**–**c**). Movement, disease, and behavioral control in suburban area only (**d**–**f**). Movement, disease, and behavioral control in rural area only (**g**–**i**). (a) Infected individuals in the large metro area. (**b**) Infected individuals in the suburban area. (**c**) Infected individuals in the rural area. (**d**) Infected individuals in the large metro area. (**e**) Infected individuals in the suburban area. (**f**) Infected individuals in the rural area. (**g**) Infected individuals in the large metro area. (**h**) Infected individuals in the suburban area. (**i**) Infected individuals in the rural area. Parameter values used are given in Table 2, Table 3 and Table 4.

**Figure 11 antibiotics-08-00081-f011:**
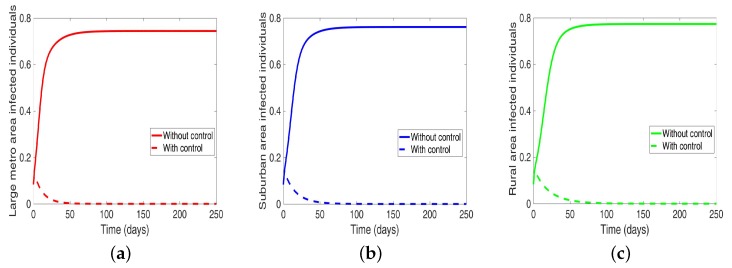
MRSA model simulation with movement, disease and behavioral control in large metro area only. (**a**) Infected individuals in large metro area. (**b**) Infected individuals in suburban area. (**c**) Infected individuals in rural area. Parameter values used are given in Table 2, Table 3 and Table 4.

**Figure 12 antibiotics-08-00081-f012:**
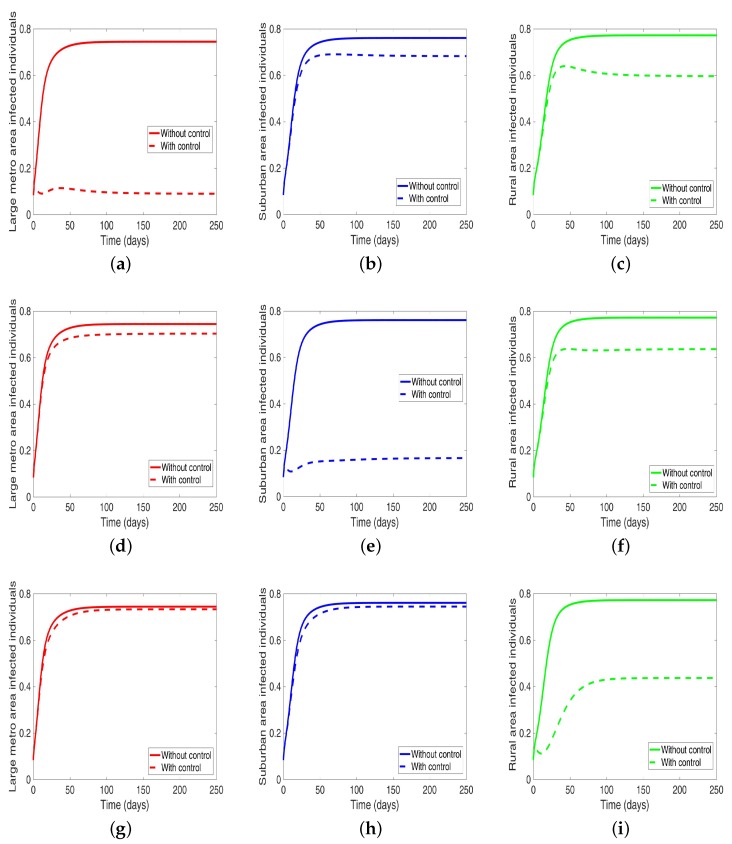
MRSA model simulation with movement, disease, and behavioral control in large metro area only (**a**–**c**). Movement, disease, and behavioral control in suburban area only (**d**–**f**). Movement, disease, and behavioral control in rural area only (**g**–**i**). (**a**) Infected individuals in large metro area. (**b**) Infected individuals in suburban area. (**c**) Infected individuals in rural area. (**d**) Infected individuals in large metro area. (**e**) Infected individuals in suburban area. (**f**) Infected individuals in rural area. (**g**) Infected individuals in large metro area. (**h**) Infected individuals in suburban area. (**i**) Infected individuals in rural area. Parameter values used are given in Table 2, Table 3 and Table 4.

**Figure 13 antibiotics-08-00081-f013:**
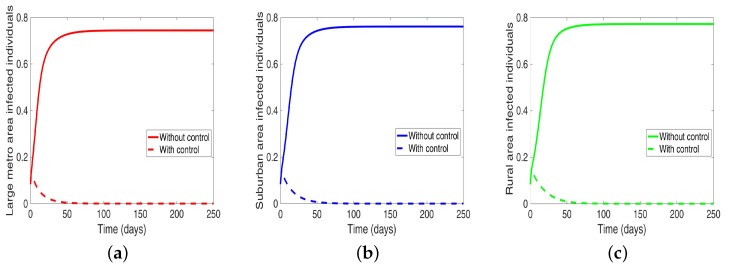
MRSA model simulation with movement, disease and behavioral control in large metro area only. (**a**) Infected individuals in large metro area. (**b**) Infected individuals in suburban area. (**c**) Infected individuals in rural area. Parameter values used are given in Table 2, Table 3 and Table 4.

**Table 1 antibiotics-08-00081-t001:** Variable and Parameter descriptions for the MRSA model Equation (Equation 1) per day.

**Variable**	**Description**
$U_{N_{i}},U_{L_{i}},U_{H_{i}}$	Population of uncolonized non-IDUs, low-risk IDUs, and high-risk IDUs in patch *i*
$C_{N_{i}},C_{L_{i}},C_{H_{i}}$	Population of colonized non-IDUs, low-risk IDUs, and high-risk IDUs in patch *i*
$I_{N_{i}},I_{L_{i}},I_{H_{i}}$	Population of infected non-IDUs, low-risk IDUs, and high-risk IDUs in patch *i*
**Parameter**	**Description**
$\pi_{N_{i}},\;\pi_{L_{i}},\;\pi_{H_{i}}$	Natural birth rate
$\beta_{N_{i}},\beta_{L_{i}},\beta_{H_{i}}$	Transmission probability per contact with colonized or infected population
$\sigma_{N_{i}},\sigma_{L_{i}},\sigma_{H_{i}}$	Progression rate of colonized population
$\gamma_{N_{i}},\gamma_{L_{i}},\gamma_{H_{i}}$	Recovery rate of infected population
$\tau_{N_{i}},\;\tau_{L_{i}},\;\tau_{H_{i}}$	Decolonization rate of colonized population
$\omega_{N_{i}},\omega_{L_{i}},\omega_{H_{i}}$	Increased IDU risk behavior
$\alpha_{N_{i}},\alpha_{L_{i}},\alpha_{H_{i}}$	Decreased IDU risk behavior
$\mu_{i}$	Natural death rate
$\delta_{N_{i}},\delta_{L_{i}},\delta_{H_{i}}$	Death rate due to disease

**Table 2 antibiotics-08-00081-t002:** Parameter estimation of the non-injection drug users reduced MRSA model given in [14] using regional data.

Variable	Large Metro	Rural	Suburb
$\beta_{N}$	0.2895	0.2015	0.1444
$\gamma_{N}$	0.2778	0.1874	0.1316
$\sigma_{N}$	0.7079	0.6670	0.6560

**Table 3 antibiotics-08-00081-t003:** Complementary table displaying and quantifying injection drug user risk associated behaviors. Each rate value (0–5) is equal to % risk factor value. (i.e., rating value of 7 = 42% risk factor).

Rating	Class Assignment	Risk Factor Limits (%)
0	non-injection drug user	0
1–6	low-risk injection drug user	6–36
7–15	high-risk injection drug user	42–90

**Table 4 antibiotics-08-00081-t004:** Census Bureau county-to-county 2009–2013 ins-outs-nets-gross number migrating in Shawnee, Douglas, and Jefferson counties [23].

Counties	Shawnee	Douglas	Jefferson
Shawnee	-	1291	578
Douglas	1291	-	511
Jefferson	578	511	-

**Table 5 antibiotics-08-00081-t005:** County-to-county 2009–2013 ins-outs-nets-gross migration rate in Shawnee, Douglas, and Jefferson counties.

Counties	Shawnee	Douglas	Jefferson
Shawnee	-	0.0074	0.0033
Douglas	0.0116	-	0.0046
Jefferson	0.0305	0.0270	-

**Table 6 antibiotics-08-00081-t006:** Proportion of infected individuals at the end of simulation period when controlling behavior in isolated patches.

Regions	Without Control	With Control
Metro	0.7440	0.4066
Suburban	0.7610	0.4137
Rural	0.7735	0.4176

**Table 7 antibiotics-08-00081-t007:** Final population size when controlling behavior one at a time when the regions are connected.

Regions	Without Control	With Control in Metro Area	With Control in Suburban Area	With Control in Rural Area
Metro	0.7442	0.4539	0.7119	0.7352
Suburban	0.7608	0.7061	0.4939	0.7486
Rural	0.7720	0.6693	0.6895	0.6230

**Table 8 antibiotics-08-00081-t008:** Final population size when controlling MRSA when the regions are Isolated.

Regions	Without Control	With Control
Metro	0.7440	0.1959
Suburban	0.7610	0.1979
Rural	0.7735	0.1924

**Table 9 antibiotics-08-00081-t009:** Final population size when controlling MRSA one at a time when the regions are connected.

Regions	Without Control	With Control in Metro Area	With Control in Suburban Area	With Control in Rural Area
Metro	0.7442	0.3095	0.7345	0.7419
Suburban	0.7608	0.7363	0.3854	0.7566
Rural	0.7720	0.6999	0.7191	0.5709

**Table 10 antibiotics-08-00081-t010:** Final population size when controlling MRSA & Behavior one ar a time when the regions are connected.

Regions	Without Control	With Control in Metro Area	With Control in Suburban Area	With Control in Rural Area
Metro	0.7442	0.0897	0.7029	0.7330
Suburban	0.7608	0.6825	0.1656	0.7446
Rural	0.7720	0.5962	0.6364	0.4366

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
