# Peer review of "Impact of Mobility on Methicillin-Resistant Staphylococcus aureus among Injection Drug Users"

_antibiotics, 2019, doi:10.3390/antibiotics8020081_

Round 1

Reviewer 1 Report

In this study, the authors developped a deterministic model for the transmission dynamics of methicillin-resistant staphylococcus aureus within the injection drug users. In the introduction, the authors exhibited in a comprehensive way the interest of study. Then, the manuscript is divided into 5 sections followed by a discussion and a conclusion.

My general comment is that the article is very dense. I suggest to reduce the two first sections, and to go quickly to the model parameter estimation. Indeed, the heart of the subject should be the section 6. Furthermore, these sections are highly similar to Wagner & Agusto 2018 (BMC Infectious Diseases), with the same figures 1, 2 and 3.

Line 81 : I would have liked that authors defined the low-risk IDUs and the high-risk IDUs. What are differences between both ? Given the model is based on these subgroups, it is very important to define them.

Line 166 – 167 : I would have find interested to present the proportion of MRSA patients according areas. Given population is bigger in large metro area, it is expected to have a highest number of patients with MRSA.  

Figure 8 legend : review the items in the figure. To simplify, I think the authors could conserve only the tables.

Line 268 – 269 : The authors could give an explication concerning the negligible effect of control in the patches with large population.

Line 292 – 293 : the authors could discuss this with the movement rates in figure 4.

Author Response

Review Report (Reviewer 1)

In this study, the authors developped a deterministic model for the transmission dynamics of methicillin-resistant staphylococcus aureus within the injection drug users. In the introduction, the authors exhibited in a comprehensive way the interest of study. Then, the manuscript is divided into 5 sections followed by a discussion and a conclusion.

My general comment is that the article is very dense. I suggest to reduce the two first sections, and to go quickly to the model parameter estimation. Indeed, the heart of the subject should be the section 6. Furthermore, these sections are highly similar to Wagner & Agusto 2018 (BMC Infectious Diseases), with the same figures 1, 2 and 3.

Response: We thank the reviewer for the suggestion, we have moved some of the earlier materials to the appendix. We have retained the model formulation section and figures 1,2, and 3 so that the paper is self-contained without the reader having to go to the Wagner & Agusto 2018 paper. Also, the inclusion of these figures gives a clear picture of why we did this current study and the source of the data.

 Line 81 : I would have liked that authors defined the low-risk IDUs and the high-risk IDUs. What are differences between both ? Given the model is based on these subgroups, it is very important to define them.

Response: We have given some description of these two subgroups in the text, and this definition follow the definition given in Wagner and Agusto.

Line 166 – 167 : I would have find interested to present the proportion of MRSA patients according areas. Given population is bigger in large metro area, it is expected to have a highest number of patients with MRSA. 

Response: We thank the reviewer for their suggestion. The numbers in Figure 2 are the data we obtained from the Agency for Healthcare and Research and Quality (AHRQ) which are not specific to the counties used in this current study. As a result we cannot present the data as proportions. However, we have used proportions to represent the populations in the manuscript, see the various figures 7 – 13 and tables 6 – 10.

Figure 8 legend : review the items in the figure. To simplify, I think the authors could conserve only the tables.

Response: We are not sure what the reviewer is suggesting we do here. The legend is to explain what the different lines are for each sub-figure. Also, the figure shows the solution profile over time while the table is at the final time. We believe both pictorial and table representation gives a clear presentation of what we are trying to say with the result we obtained/

Line 268 – 269: The authors could give an explication concerning the negligible effect of control in the patches with large population.

Response: We thank the reviewer for their suggestion we have added a sentence to explain the results further.

Line 292 – 293: the authors could discuss this with the movement rates in figure 4.

Response: We thank the reviewer for their suggestion we have added a sentence to explain the results further.

Reviewer 2 Report

I think this is a very good written paper, and it represents a remarkable example of mathematical modelization in public health and how this could be useful in research. The presented model, previously described in another paper, seems to be a sound foundation on which new studies may be planned in the future.

I would suggest the following corrections:

Line 103: probably a typo: substitute “no” with “not”

Lines 174-180: The estimation of parameters εL and εH are well described pointing to a previous publication. I would also suggest to include the possibility of estimating the severity of addiction and therefore the classification of low- and high-risk IDU, from clinical records, as reported for example in:

C. Quercioli, G. Messina, P. Fini, C. Frola, e N. Nante, «Is it possible to evaluate addiction from clinical records? Testing a retrospective addiction severity evaluation measure», Subst Use Misuse, vol. 45, n. 12, pagg. 2045–2058, ott. 2010.

Line 185: I would consider to state explicitly how sizes are defined according to the Department of Human Services, to compare them with the studied counties

Line 193: probably a typo: substitute “countries” with “counties”

Line 216: probably a typo: substitute “carries” with “carriers”

Line 347: probably a typo: substitute “compare” with “compared”

Lines 398 and 400: probably a typo: substitute “inventions” with “interventions”

Author Response

Review Report (Reviewer 2)

I think this is a very good written paper, and it represents a remarkable example of mathematical modelization in public health and how this could be useful in research. The presented model, previously described in another paper, seems to be a sound foundation on which new studies may be planned in the future.

Response: We thank the reviewer for their kind words.

I would suggest the following corrections:

Line 103: probably a typo: substitute “no” with “not”

Response: We have corrected the typo.

Lines 174-180: The estimation of parameters εL and εH are well described pointing to a previous publication. I would also suggest to include the possibility of estimating the severity of addiction and therefore the classification of low- and high-risk IDU, from clinical records, as reported for example in:

C. Quercioli, G. Messina, P. Fini, C. Frola, e N. Nante, «Is it possible to evaluate addiction from clinical records? Testing a retrospective addiction severity evaluation measure», Subst Use Misuse, vol. 45, n. 12, pagg. 2045–2058, ott. 2010.

Response: We thank the reviewer for the suggestion, however, the authors in the suggested literature used a different metric than was used in the Wagner and Agusto paper which determined the risk level solely on drug use rather than addiction. Although, the Wagner and Agusto metric implicitly relied on some of the metric used for the addiction metric such as physical health, drug use, family/social relationships, psychological health. It will be good to consider other metric as well and we are open to doing so in future studies. We have included a sentence about the possibility of using addiction severity score to determine the risk factors.

Line 185: I would consider to state explicitly how sizes are defined according to the Department of Human Services, to compare them with the studied counties

Response: We thank the reviewer for the suggestion. We meant Department of Health and Human Services and we had already stated the definition of population size by the department in the introduction section. We believe that if we use the same size of the population defined by Department of Health and Human Services the conclusion from our study will be similar as the result we have in this current. And we do not see the need to compare the sizes of the areas other than what we have said on this line.

Line 193: probably a typo: substitute “countries” with “counties”

Response: We have corrected the typo.

Line 216: probably a typo: substitute “carries” with “carriers”

Response: We have corrected the typo.

Line 347: probably a typo: substitute “compare” with “compared”

Response: We have corrected the typo.

Lines 398 and 400: probably a typo: substitute “inventions” with “interventions”

Response: We have corrected the typo.
